# The Role of Non-Coding RNAs in the Neuroprotective Effects of Glutathione

**DOI:** 10.3390/ijms22084245

**Published:** 2021-04-19

**Authors:** Chisato Kinoshita, Koji Aoyama

**Affiliations:** Department of Pharmacology, Teikyo University School of Medicine, 2-11-1 Kaga, Itabashi, Tokyo 173-8605, Japan

**Keywords:** glutathione, antioxidant, neuroprotection, long non-coding RNA, microRNA, circular RNA, oxidative stress, central nervous system

## Abstract

The establishment of antioxidative defense systems might have been mandatory for most living beings with aerobic metabolisms, because oxygen consumption produces adverse byproducts known as reactive oxygen species (ROS). The brain is especially vulnerable to the effect of ROS, since the brain has large amounts of unsaturated fatty acids, which are a target of lipid oxidation, as well as comparably high-energy consumption compared to other organs that results in ROS release from mitochondria. Thus, dysregulation of the synthesis and/or metabolism of antioxidants—particularly glutathione (GSH), which is one of the most important antioxidants in the human body—caused oxidative stress states that resulted in critical diseases, including neurodegenerative diseases in the brain. GSH plays crucial roles not only as an antioxidant but also as an enzyme cofactor, cysteine storage form, the major redox buffer, and a neuromodulator in the central nervous system. The levels of GSH are precisely regulated by uptake systems for GSH precursors as well as GSH biosynthesis and metabolism. The rapid advance of RNA sequencing technologies has contributed to the discovery of numerous non-coding RNAs with a wide range of functions. Recent lines of evidence show that several types of non-coding RNAs, including microRNA, long non-coding RNA and circular RNA, are abundantly expressed in the brain, and their activation or inhibition could contribute to neuroprotection through the regulation of GSH synthesis and/or metabolism. Interestingly, these non-coding RNAs play key roles in gene regulation and growing evidence indicates that non-coding RNAs interact with each other and are co-regulated. In this review, we focus on how the non-coding RNAs modulate the level of GSH and modify the oxidative stress states in the brain.

## 1. Introduction

The theory on the origin of life was first proposed by Russian biochemist Aleksandr Ivanovich Oparin [1]. According to his theory, the biological systems of the most primitive organisms may be much simpler than those of current life on Earth [2]. As the subsequent RNA World concept explained, RNA or RNA-like chemicals were likely to carry out most of the information processing and metabolic transformations needed for biology to emerge from chemistry in the early history of life [3,4]. This is consistent with the surprising finding in Tetrahymena that catalytic RNAs—i.e., ribozymes—carry out enzymatic functions similar to those carried out by proteins [5]. RNA can play several roles, including copying DNA and synthesizing proteins, functioning as a structural component of ribosomes and ribozymes and regulating various cellular processes [6]. It has long been considered that non-coding RNAs that cannot translate into a protein product are merely “junk” RNA [7]. However, such non-coding RNAs have become a hot topic in recent research [8]. It is increasingly becoming clear that many non-coding RNAs are in fact highly functional in biological systems and compensate for their inability to be translated into proteins through alternate mechanisms [9].

The evolution of primitive organisms might have been destined to proceed quite gradually in the primitive ocean. However, with the appearance of the first plants, known as cyanobacteria, and the initiation of photosynthesis by these organisms, the explosive evolution that is characteristic of life on this planet was begun [10]. On the other hand, adverse effects may have driven many of the primitive organisms to extinction because of the explosive increase of atmospheric oxygen [11]. The system of oxygen consumption thus contributed greatly to the evolution of biological species, but it also generated byproducts ROS that are highly toxic to living cells [12]. To protect themselves from ROS attack, surviving living beings internally acquired several distinct antioxidative defense systems [13,14]. Assuming that RNA was major molecules that regulate biological systems of life on Earth at that time, the fact that RNA can be easily changed or mutated might have made it ideal for rapid protective action against sudden ROS attack. In this regard, it is perhaps not surprising that several types of RNA make up some portions of antioxidant systems.

Neuroprotection against ROS attack in the central nervous system (CNS) is one of the most important goals of the antioxidant systems in animals with aerobic metabolisms, since the brain governs all organs of the body, including itself. In addition, the brain consumes more oxygen per unit of its weight than any other organ and contains numerous fatty acids that could be ROS targets for oxidization [15,16]. GSH is especially important in the brain, as evidenced by the fact that the levels of GSH in the brain are higher than the levels of other antioxidants, including catalase, superoxide dismutase (SOD) and glutathione peroxidase (GPx) [17,18]. An excess of ROS and/or depletion of antioxidants, which are defined as oxidative stress states, causes a number of clinically important diseases, including neurodegenerative diseases (NDs) [19]. Recently, several non-coding RNA studies in humans also revealed that the down-regulation and/or overexpression of non-coding RNAs is linked to diseases closely related to oxidative stress [20]. In this review, we focus on the regulatory mechanism of GSH neuroprotection by non-coding RNAs.

## 2. Antioxidants against Oxidative Stress

For the proper physiological activities of living beings, a balance is required between the levels of oxidants and antioxidants. An imbalance of redox states caused by an excess of oxidants, a depletion of antioxidants or both is defined as an oxidative stress state [21]. Oxidative stress has been implicated in the etiology of various diseases, such as NDs, cardiovascular diseases, chronic obstructive pulmonary disease, chronic kidney disease, obesity, and cancer [22]. On the other hand, ROS act as signaling molecules and play important roles in a variety of physiological functions, including the regulation of autophagy, immunity, and differentiation [23,24]. ROS are generated during the mitochondrial electron transport of aerobic respiration, as well as during cellular responses to xenobiotics, cytokines, and bacterial invasion [25]. ROS is a general term that includes molecules or ions formed by highly reactive and partially reduced oxygen metabolites [26].

### 2.1. Glutathione

One of the most important antioxidants is GSH, which is a tripeptide composed of three amino acids, cysteine, glutamate and glycine [27]. GSH is the major non-protein thiol distributed in most cells, and also functions as a storage and transport form of cysteine and an important player in antioxidative defense [27]. An excess amount of cysteine can be toxic to the cells because it induces free radical generation and extracellular glutamate production [28,29]. In addition, it has recently been reported that cysteine impairs mitochondrial respiration by limiting iron bioavailability through an oxidant-based mechanism [30]. GSH is a non-enzymatic antioxidant that acts as an impregnable defense against all forms of ROS [17]. In the brain, GSH is extremely important because the brain contains an abundance of lipids with unsaturated fatty acids that act as a source of peroxidation [31].

The biosynthesis of GSH occurs via a two-step ATP-requiring enzymatic process that is catalyzed by glutamate-cysteine ligase (GCL; also known as γ-glutamylcysteine synthetase) and glutathione synthetase (GSS) (Figure 1) [17]. GCL catalyzes the formation of a dipeptide, γ-glutamylcysteine (γ-GluCys), from glutamate and cysteine, which is the rate-limiting step in GSH biosynthesis [32]. GCL is composed of a catalytic subunit (GCLc) and modulatory subunit (GCLm) [32,33]. GCLc possesses the enzymatic activity, while GCLm controls the kinetics of GCLc activity for GSH [34]. Mice with knockout of GCLm had as much as 70% depletion of brain GSH, which manifested only as mild abnormalities in neurons [35], while GCLc-knockout mice were embryonic lethal, and its conditional knockout of GCLc shows critical GSH depletion following neurodegeneration [36]. GSS is the enzyme in charge of the second step in GSH biosynthesis, which couples γ-GluCys with glycine to generate GSH [37]. Mice homozygous for GSS knockout died before embryonic day 7.5, while heterozygous mice survived with no distinct phenotype, probably because their GSH levels remained intact [38]. Cases of GCL or GSS deficiency in humans have been reported; these deficiencies are autosomal recessive metabolic disorders causing impaired physiological functions such as neuronal dysfunction and can cause mortality in early life [39].

Once GSH acts as an electron donor, a disulfide bond is formed to produce oxidized glutathione (GSSG) [40]. GSSG is a substrate of the flavoenzyme glutathione reductase (GSR), which transfers an electron from nicotinamide adenine dinucleotide phosphate (NADPH) to GSSG, thereby regenerating GSH and establishing a system for recycling GSH [41]. On the other hand, glutathione-*S*-conjugation by glutathione-*S*-transferases (GST) consumes GSH [42]. GSH-conjugated compounds are actively pumped out of the cell by numerous members of the multidrug resistance-associated protein (MRP) family [43,44,45,46].

### 2.2. Antioxidant Enzymes

SOD and catalase as well as GPx are enzymes that have a protective effect against oxidative stress (Figure 1) [47]. SOD plays a role in converting superoxide anions to less noxious hydrogen peroxide [48]. Both catalase and GPx effectively disproportionate hydrogen peroxide into harmless water and oxygen [49,50]. While catalase does not require any activator, GPx requires GSH for its activity as an electron donor. The formation of a disulfide bond between two GSH molecules gives rise to GSSG in the GPx-catalyzed reaction [50].

Three types of SOD have been identified. SOD2 (also known as MnSOD) is mainly expressed in the mitochondria and requires manganese ion for its activation. SOD1 (also known as Cu/ZnSOD) and SOD3 (also known as ECSOD), which require copper and zinc ion, are respectively expressed intracellularly and extracellularly [51,52]. It has been well established that mutations of the SOD1 gene are implicated in the etiology of amyotrophic lateral sclerosis (ALS), one of the NDs [53]. A hallmark of SOD1-associated ALS is the deposition of SOD1 into insoluble aggregates in motor neurons, probably as a result of mutation-induced structural destabilization and/or oxidative damage because of mutation, which in turn contributes to the misfolding and aggregation of SOD1 into neurotoxic species [54].

Catalase is a heme-containing homotetrameric protein without any isoforms. Mice lacking catalase develop normally, although catalase deficiency or mutation in humans results in acatalasemia, which is characterized by oral gangrene, altered lipid, carbohydrate, and homocysteine metabolism and an increased risk of diabetes mellitus, although there are only limited reports of acatalasemia in the literature [55,56,57,58].

Eight isoforms of GPx have been identified, GPx1 to GPx8 [59]. GPx1-4 are selenium-dependent enzymes, whereas GPx5, GPx7 and GPx8 are selenium-independent but contain a cysteine instead of a selenocysteine [50]. GPx1 is the most abundant cellular GPx and functions as an important antioxidative enzyme that interacts with fatty acid hydroperoxides as well as hydrogen peroxide in the brain. On the other hand, GPx2 and GPx3 have been identified as gastrointestinal and plasma GPx isoforms, respectively. GPx4 is a ubiquitously expressed peroxidase that can directly reduce lipid hydroperoxides in the cellular membrane [60]. GPx5 is an epididymal-specific secretory GPx, and GPx6 is expressed in the olfactory epithelium. GPx7 and GPx8 have been observed in the lumen and membrane of the endoplasmic reticulum (ER), respectively. Studies using GPx1- knockout mice showed that these animals were phenotypically normal, but they were particularly susceptible to oxidative stress in the brain [61,62,63]. Although homozygotes of GPx4-knockout mice are embryonic lethal, heterozygous mice do not show any alterations in the activities of other major antioxidant defense enzymes, such as Gpx1 and catalase [64,65]. However, it has been determined that Gpx4 heterozygotes show increased lipid peroxidation in the brain [66].

### 2.3. Glutaredoxin, Thioredoxin and Peroxiredoxin

Glutaredoxins (Grxs), thioredoxins (Trxs) and peroxiredoxins (Prxs) have been characterized as electron donors, and shown to function in the protection of the intracellular redox state and as antioxidants [67,68]. Both Grxs and Trxs are members of a superfamily of low-molecular-mass proteins that catalyze the reduction of disulfide bonds in a variety of proteins. GSSG formed in the Grx reaction is reduced by GSR at the expense of NADPH, whereas oxidized Trx is reduced by thioredoxin reductase (TrxR) with electrons transferred from NADPH [69]. Prxs are peroxidase enzymes that receive electrons from NADPH by coupling with Trx and TrxR [70].

### 2.4. The Nrf2-Keap1 System

Nuclear factor erythroid 2-related factor 2 (Nrf2) is one of the most important transcriptional factors, with a responsibility for regulating hundreds of antioxidant genes involved in the synthesis, metabolism and conjugation of GSH [71]. Kelch-like ECH-associated protein1 (Keap1) has been identified as a factor that negatively regulates Nrf2. Under normal conditions, Keap1, which forms a ubiquitin E3 ligase complex with Cullin3, binds to Nrf2, trapping Nrf2 into the cytosol and promoting its ubiquitination and proteasomal degradation. On the other hand, oxidative stress conditions modify the cysteine residues of Keap1, facilitating the dissociation of Nrf2 from Keap1, which promotes the nuclear translocation of Nrf2. In the nucleus, Nrf2 heterodimerizes with small musculo-aponeurotic fibrosarcoma (sMAF) and interacts with antioxidant response elements (ARE) in the promoter region of target genes, resulting in their transcriptional activation [72]. Krüppel-like factor 2 (KLF2) is a member of the zinc finger transcription factor family and acts to prime Nrf2 activation through enhanced nuclear localization [73]. Nrf2 transcriptionally regulates several antioxidant genes, including GCLc, GCLm, GSS, GPx4, GSR, Trx1, TrxR and catalase, some of which are also up-regulated by KLF2 [74]. Nrf2-knockout mice lose the ability to induce antioxidative genes such as GPx, SOD, GST, and catalase, with the result that the GSH system for protection against oxidants is not induced, and the mice are left vulnerable to oxidative stress [75,76].

## 3. Uptake System for the Sources of GSH

### 3.1. Cysteine Uptake System

Although GSH is composed of three amino acids, the determinant substrate for neuronal GSH synthesis is cysteine [37]. Cysteine uptake in neurons is mostly mediated by sodium-dependent systems, mainly the excitatory amino acid carrier 1 (EAAC1; also known as EAAT3 or SLC1A1) (Figure 1) [77]. EAAC1 is one of the five excitatory amino acid transporters (EAATs) that are collectively known as solute carrier family 1 (SLC1). EAAC1-deficient mice exhibit an approximately 40% decrease in brain GSH content and neurodegeneration in advanced age [77]. Further, overexpression of Nrf2 in brain neurons is sufficient to upregulate both neuronal EAAC1 protein and GSH content, and these effects were abrogated in mice genetically deficient in either Nrf2 or EAAC1 [78]. EAATs are responsible for the uptake of glutamate; the known EAATs include glutamate aspartate transporter (GLAST; also known as EAAT1 or SLC1A3), glutamate transporter-1 (GLT-1; also known as EAAT2 or SLC1A2), EAAC1, EAAT4 (also known as SLC1A6) and EAAT5 (also known as SLC1A7) [79]. The EAATs are secondary active transporters, translocating three sodium ions and one proton and counter-transporting one potassium ion for each substrate, and thereby supplying the energetic driving force to transport glutamate against its electrochemical gradient [80]. Recent lines of evidence show that cysteine transport through EAAC1 is facilitated through cysteine deprotonation and that, once inside, the thiolate is rapidly reprotonated [81,82]. GLAST and GLT-1 are expressed in astrocytes, whereas EAAC1, EAAT4 and EAAT5 are expressed in neurons [83]. EAAC1 and GLT-1 are widely distributed throughout the brain, although the distribution of GLAST, EAAT4 and EAAT5 is restricted.

### 3.2. Cystine Uptake System

In the extracellular environment, cysteine is mainly present in its oxidized form, cystine [84]. Cystine is intracellularly reduced to cysteine once imported into the cell, and turns out to be a building block of the antioxidant GSH [85]. Cystine transport is mediated via system xc-, which is composed of a heavy chain subunit 4F2hc (encoded by the *SLC3A2* gene) and a light chain specific subunit xCT (encoded by the *SLC7A11* gene). System xc- exchanges glutamate for cystine in a 1:1 ratio according to the respective concentration gradients. Investigation into the localization of system xc- at the cellular level in the brain using immunohistochemistry in rat brain slices showed that system xc- is localized mainly in astrocytes, but not in neurons [86]. In addition, the inflammatory stimulation specifically upregulates system xc- activity in astrocytes but not in microglia and neurons [86,87,88,89]. However, another study using immunohistochemistry showed that xCT was localized in both neurons and astrocytes in the mouse and human brain [90]. Since deficiency of EAAC1 in mice results in a significant decrease in brain GSH content but not zero brain GSH content [77], system xc- may also play a small role in uptake of the sources of GSH in neurons.

Ferroptosis is a recently described form of non-apoptotic regulated cell death caused by iron-dependent lipid peroxidation that is distinct in its morphological and genetic profile from other cell death mechanisms such as apoptosis, necroptosis, or autophagy [91]. GPx4 was identified as a key regulatory factor in ferroptosis, the process of detoxifying membrane lipid peroxidation by converting lipid peroxides to non-toxic lipids [92,93]. GSH is a cofactor for GPx4 and is required for the detoxification of lipid peroxidation [94]. Inhibition of system xc- caused cysteine depletion, resulting in a lack of GSH synthetic substrate, leading to impairment of GPx4 function and finally ferroptosis [95,96]. Further, GPx4 inhibition via Ras-selective lethal small molecule 3 or genetic knockdown of GPx4 could also induce ferroptosis [96].

## 4. Neuroprotective Function of Non-Coding RNA

Soon after DNA was determined to be the store of genetic information in the eukaryotic nucleus, and after proteins were shown to be synthesized in the cytoplasm based on this information, RNA was first recognized in the form of the messenger (mRNA) that passed genetic information from the DNA to the protein synthetic machinery as explained in the classical central dogma of molecular biology [97]. Then, the functional RNAs, i.e., transfer RNA (tRNA) and ribosomal RNA (rRNA), were found to be involved in protein synthesis. Sometime later, several small non-mRNAs, other than rRNA and tRNA, were detected and isolated with associating ribonucleoprotein (RNP) complexes. These short non-coding RNAs play an essential role in the maturation of functional RNAs, which are small nuclear RNAs (snRNAs) involved in splicing events and small nucleolar RNAs (snoRNAs) guiding posttranscriptional modifications on rRNAs and snRNAs [97,98].

Recently, several classes of regulatory non-coding RNAs, including microRNAs, long non-coding RNAs, circular RNAs, and so on, have been discovered to act as key regulators of gene expression in many different cellular pathways and systems [99]. These different types of non-coding RNAs are found to confer neuroprotection against oxidative stress by linking with each other [20].

### 4.1. MicroRNAs

MicroRNAs (miRNAs), which are a class of short non-coding RNAs approximately 20 nucleotides in length, are among the most well-studied ncRNAs [100]. The function of miRNAs is mainly to silence target expressions by binding to target gene transcripts located mainly at the 3′-untranslated regions (3′-UTR). Clustered miRNAs can either be simultaneously transcribed from single polycistronic transcripts containing multiple miRNAs or independently transcribed. In most cases, RNA polymerase II transcribes the primary miRNAs (pri-miRNAs), which are cleaved by a complex called a microprocessor, which contains the ribonuclease III Drosha and the RNA-binding protein DGCR8/Pasha, to generate small hairpin-shaped RNAs, called miRNA precursors (pre-miRNAs). Pre-miRNAs are exported by exportin-5 in complex with RAN-GTP and are processed by a double-stranded ribonuclease III enzyme termed Dicer, which is complexed with TRBP. The mature miRNA duplexes are loaded onto an Argonaute protein to form an effector complex called the RNA-induced silencing complex (RISC). Finally, one strand of the miRNA is removed from RISC to generate the mature RISC that induces gene silencing. The post-transcriptional regulation by the RISC complex is mediated by incomplete base-paring of miRNA-mRNA interactions, likely due to the targeting of multiple transcripts, which contributes to the complexity or redundancy of miRNA systems.

Numerous miRNAs have been identified as regulators of GSH-regulating factors. Several reports have shown that GSH or ROS levels are modulated by miRNAs through the regulation of factors related to GSH synthesis and/or metabolism in the brain or neuronal cells. Among them, the Nrf2-mediated pathway has been well-studied in relation to the regulation of neuroprotection and oxidative stress (Figure 2). MiR-23a-3p targets Nrf2 itself, and injection of antagomiR into the lateral ventricle results in an increase of GPx4 expression and an inhibition of ROS accumulation and lipid peroxidation in a rat model of intracerebral hemorrhage [101]. In addition, hydrogen peroxide-responsive miR-153 directly targets and inhibits gene expression of Nrf2, which results in ROS accumulation by preventing transactivation of the downstream antioxidative gene GCLc [102]. On the other hand, miR-7 can activate the Nrf2 pathway by directly targeting the 3′-UTR of Keap1 mRNA to increase GSH levels by inducing an increase in GCLm expression [103]. Nrf2 activator KLF2 is targeted by miR-25, and this action results in an inhibition of proliferation while promoting the apoptosis of hippocampal neurons with a reduction of GSH level and antioxidative enzyme expression [104]. It has also been reported that the inhibition of miR-592 function promotes an increase in GSH levels and a decrease of ROS via an Nrf2-signaling pathway by up-regulating KIAA0319, which is a dyslexia-associated protein [105,106]. An Nrf2-signaling pathway is also modulated by miR-139 or miR-144, both of which have been shown to modulate redox states, although the precise mechanisms by which they modulate the Nrf2-signaling pathway remain unknown [107,108]. The PI3K/Akt pathway plays key roles in regulating Nrf2-dependent protection against oxidative stress (Figure 2). Downregulation of miR-200c-3p increases GSH and SOD levels and reduces the damage to hippocampal neurons by upregulating the reversion-inducing-cysteine-rich protein with kazal motifs (RECK) and inactivating the Akt signaling [109]. Inhibition of miR-204-5p alleviated oxidative injuries in hippocampal neuronal cells via a reduction of ROS and oxidative stress marker malondialdehyde (MDA) levels and an upregulation of SOD and GSH, possibly by targeting brain-derived neurotrophic factor (BDNF) in the neurotrophic tyrosine kinase receptor type 2 (TrkB)-mediated pathway [110]. Upregulation of miR-409 expression activates the Akt-regulated glycogen synthase kinase 3β (GSK3β), leading to an increase in GSH and SOD levels and decrease in ROS levels, which protects against ROS-induced neurotoxicity [111]. MiR-214 plays a neuroprotective role characterized by an increase of SOD and GSH levels directly targeting phosphatase and tensin homolog deleted from chromosome 10 (PTEN), a suppressor of Akt signaling [112]. The mitogen-activated protein kinase (MAPK) pathway is also important because MAPKs such as ERK, p38MAPK and JNK are downstream effectors of antioxidant responses and changes in GSH levels (Figure 2). MiR-410 promotes decreased MDA content but increased SOD activities and GPx activities in hippocampal neurons by regulating the p38/JNK pathway through the inhibition of tissue inhibitors of metalloproteinase 2 (TIMP2) [113]. Induction of miR-486 by ROS generation inhibits NeuroD6 expression in the p38/JNK pathway, resulting in ROS accumulation via downregulation of antioxidative genes including GPx3 [114]. The upregulation of miR-136 could potentially inhibit inducible nitric oxide synthase (iNOS) activation as well as the apoptosis of neurons by negatively targeting Kallikrein-related peptidase 7 (KLK7) through inhibition of the p38 signaling pathway, resulting in increased content of SOD and GPx, as well as reduced MDA content [115]. Other signaling pathways are also cross-linked and interact with each other to regulate intracellular GSH levels. MiR-146a has a protective effect in the brain by repressing the tumor necrosis factor receptor associated factor (TRAF6)-mediated nuclear factor kappa B (NF-κB) pathway, leading to inhibition of inflammation and oxidative stress [116]. MiR-98 improves oxidative stress and mitochondrial dysfunction through activation of the Notch signaling pathway by binding to Hairy/enhancer-of-split related with YRPW motif protein 2 (HEY2) with enhancement of viability in hippocampal neurons [117]. Overexpression of miR-129-3p can alleviate oxidative stress and ROS-mediated apoptosis of hippocampal neurons by targeting mitochondrial calcium uniporter (MCU) [118]. The suppression of miR-200a can inhibit apoptosis in striatal neuron cells, increase the levels of GSH and SOD and decrease the level of MDA in the brain tissue by upregulating dopamine receptor D2, and thereby, repressing the cAMP/PKA signaling pathway [119]. MiR-320 affects cellular proliferation, apoptosis, and oxidative stress by inhibiting the NADPH oxidase 2 (Nox2) pathway [120]. EAAC1 is negatively regulated by miR-96-5p, and blocking of miR-96-5p by the administration of an inhibitor increased the levels of EAAC1 and GSH and had a neuroprotective effect against oxidative stress in the mouse substantia nigra [121]. In addition, miR-96-5p can indirectly regulates GTRAP3-18, a negative regulator of EAAC1, through directly targeting RNA-binding protein NOVA1 and modulate the levels of GSH in the mouse dentate gyrus of hippocampus as well as SH-SY5Y cells [122]. Multidrug resistance-associated protein 1 (MRP1), which is a direct target of miR-199a-5p, plays a key role in clearing intracellular GSSG [123]. These reports indicate that up- or down-regulation of miRNA function can modulate GSH levels and thereby regulate the neuroprotective effects of GSH (Table 1).

### 4.2. Long Non-Coding RNAs

Long non-coding RNAs (lncRNAs) are long RNA transcripts that are not protein-coding, and are longer than 200 nucleotides by definition [124]. The transcription for lncRNAs is transcribed by RNA polymerase II, similar to the transcription for mRNAs, as they are capped, spliced and polyadenylated, however, lncRNAs lack a translated open reading frame. LncRNA is a quite heterogenous term, in the sense that it encompasses several functionally different types of lncRNAs that have different biogenetic mechanisms. LncRNAs are roughly classified based on their position relative to protein-coding genes: exon or intron sense-overlapping, intergenic, antisense, bidirectional and enhancer lncRNAs [124]. Functionally, lncRNAs can guide transcription factors to specific genomic locations for the regulation of gene expression, work as a scaffold to facilitate the assembly of chromatin remodeling complexes, serve as a sponge to titrate miRNAs out from their mRNA targets, or bind to transcription factors or other proteins as a decoy and sequester them away from chromatin [124]. Some lncRNAs play a role in neuroprotection mostly by modulating the expression of miRNAs by acting as their sponge [20] (Figure 3).

Nuclear paraspeckle assembly transcript 1 (NEAT1, an lncRNA) is transcribed from familial tumor syndrome multiple endocrine neoplasia (MEN) type 1 and encodes two transcriptional variants. NEAT1 acts as a sponge for miR-1277-5p targeting Rho GTPase activating protein 26 (ARHGAP26), resulting in oxidative stress with a reduction of GPx and SOD activities in cells of the neuroblastoma cell line, SK-N-SH [125]. LncRNA plasmacytoma variant translocation 1 (PVT1) is an intergenic lncRNA with multiple splice isoforms. PVT1 has been reported to regulate ferroptosis through miR-214-mediated p53, upregulation of which markedly reduced the expressions of xCT and GPx4 in SK-N-SH cells [126]. LncRNA AK046177 appears to be transcribed from the loci at the protein encoding region of poly (ADP-ribose) polymerase family, member 8 (Parp 8). Downregulations of AK046177 and miR-134, along with increasing intracellular cAMP levels and activation of the Nrf2 pathway leading to reductions of ROS and increases of SOD and GPx activities thereby protecting the brain [127]. LncRNA H19 is a maternally expressed and paternally imprinted gene located near the telomeric region adjacent to the insulin like growth factor 2 (IGF2) gene [128]. H19 acts as a sponge for miR-148a-3p, which could target Rho associated coiled-coil containing protein kinase 2 (ROCK2), such that inhibition of H19 has a neuroprotective effect to increase SOD and GPx activities while decreasing ROS level [129]. There is another report about lncRNA H19 which shows that downregulation of H19 suppresses hippocampal neuron apoptosis by inhibiting IGF2 methylation [130]. When the lncRNA gene is an antisense gene that overlaps the genomic coordinates of a protein-coding gene on the opposite strand, the lncRNA genes are named as protein-coding gene symbol with the suffix -AS and sequential number. β-secretase 1 antisense RNA (BACE1-AS) downregulates its antisense coding protein BACE1 by down-regulation of miR-34b-5p, and down-regulation of BACE1-AS improves dopamine-dependent oxidative stress by inhibition of iNOS activation in the substantia nigra [131]. Further, Wilms’ tumor 1 antisense RNA (WT1-AS) suppresses the expression of WT1 by acting as an miR-375 sponge and plays a neuroprotective role in SH-SY5Y cells [132]. Silencing of lncRNA SRY-Box transcription factor 21 (SOX21) antisense divergent transcript 1 (SOX21-AS1) could act to alleviate neuronal oxidative stress and suppress neuronal apoptosis through the upregulation of Frizzled3/5 (FZD3/5) and subsequent activation of the Wnt signaling pathway in hippocampal neurons [133]. These results suggest that activation or inactivation of lncRNA contributes to neuroprotection mainly by allowing lncRNA to act as a functional miRNA sponge (Table 2).

### 4.3. Circular RNAs

Circular RNAs (circRNAs) are single-stranded RNAs that form circular molecules in which the 3′ and 5′ ends are covalently linked [134]. The classic regulatory mechanisms of circRNAs involve their role as competitive endogenous RNAs (ceRNAs). CircRNAs can regulate gene expression by influencing transcription, mRNA turnover, and translation by sponging RNA-binding proteins and miRNAs. A search on PubMed uncovered no reports of specific circRNAs modulating the levels of GSH or oxidative stress specifically in the brain or neuronal cells. However, most of the miRNAs that can modulate GSH levels, which are listed in Table 1, have been reported to be regulated by several circRNAs. For example, CDR1as (also named ciRS-7 or CDR1NAT), which is formed by reverse splicing of the antisense strand of the cerebellar degeneration-associated antigen 1 (CDR1) gene [135], has more than 70 miR-7-binding sites that inhibit its binding to target genes [134]. As described above, miR-7 targets Keap1 to regulate and modulate the neuroprotective effects of GSH [103].

There are some reports that have shown that circRNAs modulate protective effects against cellular damage, although these effects are not related to neurons or the brain (Figure 4). Knockdown of circRNA_0084043 remarkably reduced oxidative stress as evidenced by the down-regulated MDA content, enhanced activities of SOD and GPx via sponging of miR-140-3p and regulation of transforming growth factor alpha (TGFA) in a hyperglycemia-induced human retinal pigment epithelial cell, ARPE-19 [136]. Up-regulation of circHIPK3 or down-regulation of circHIPK3 targeting miR-221-3p mediated the promotion of proliferation, inhibition of apoptosis, decrease of MDA level and increase of GPx level in human lymphatic endothelial cells [137]. CircHIPK3 also inhibits proliferation and induces apoptosis of cardiomyocytes via binding to miRNA-124-3p in human cardiac myocytes (HCM) [138]. CircRNA_0001445 inhibits oxidized LDL-induced inflammation, oxidative stress and apoptosis by regulating miRNA-640 in human umbilical vein endothelial cells (HUVECs) [139]. CircIL4R acts as a miR-541-3p sponge to regulate its target GPx4, the upregulation of which relieved the miR-541-3p-induced tumor inhibition and ferroptosis aggravation in hepatocellular carcinoma [140]. CircEPSTI1 attenuates the effects of ferritin, which are mediated by xCT, which in turn regulates lipid peroxidation and GSH levels in cervical cancer [141] (Table 3). Further research will shine light on the contribution of circRNAs to the neuroprotective effects of GSH.

## 5. Conclusions

A novel class of non-coding RNAs is increasingly being identified along with the evolution of RNA technologies. These RNAs appear to regulate gene expression by linking and interacting with each other. Recently, abnormalities in the expression of these non-coding RNAs have been reported to be involved in the cause and/or progress of several diseases, including NDs. As described in this review, several types of non-coding RNAs have been reported to contribute to the regulation of GSH synthesis and/or metabolism in various cell lines and tissues. Since GSH depletion has long been observed in patient with NDs, it may be that the etiology of NDs involves oxidative stress induced by alterations in the expression of non-coding RNAs. This is a meaningful indication that modulation of the expression of non-coding RNAs could be a novel therapeutic approach for NDs via an increase in GSH levels. Recently, some of the miRNA-based therapeutics are actually processed in preclinical and clinical trial for various diseases including NDs. Understanding the interplay of miRNAs and other non-coding RNAs is key for the development of next generation drugs for the cure of NDs.

## Figures and Tables

**Figure 1 ijms-22-04245-f001:**
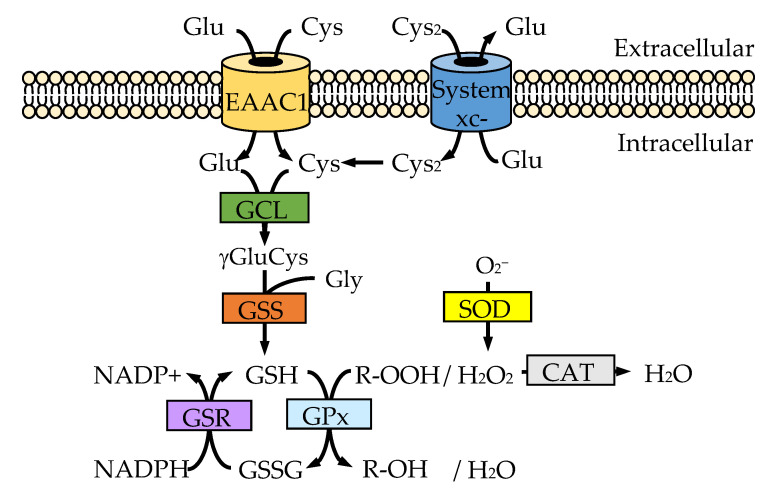
Regulation of the redox system. Among the three amino acids that form GSH—i.e., cysteine (Cys), glutamate (Glu) and glycine (Gly)—Cys is the rate-limiting substrate. Cys is supplied via the Cys transporter EAAC1 or cystine (Cys_2_) transporter system xc-. Cys_2_ is intracellularly reduced to Cys once imported into the cell, and turns out to be a building block of the antioxidant GSH.GSH synthesis is catalyzed by GCL and GSS. GSR transfers an electron from nicotinamide adenine dinucleotide phosphate (NADPH) to GSSG, and thereby catalyzes the reduction of GSSG to GSH. GPx reduces peroxide (R-OOH) to a harmless compound (R-OH) by gathering the needed reducing equivalents from GSH. SOD converts superoxide anion to less noxious hydrogen peroxide (H_2_O_2_), and catalase (CAT) reduces H_2_O_2_ without any activator.

**Figure 2 ijms-22-04245-f002:**
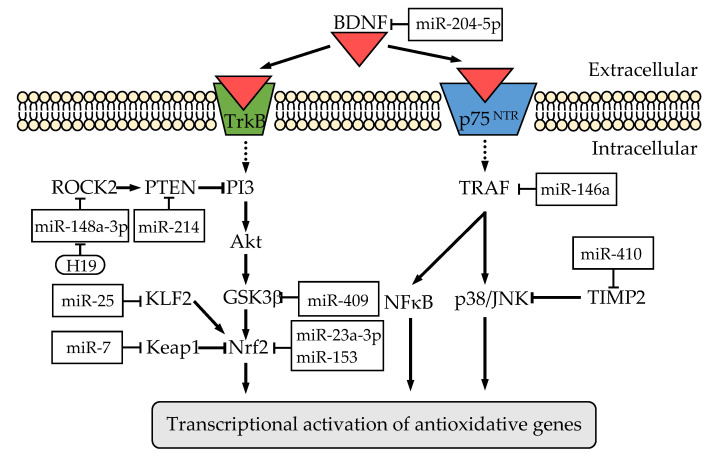
Signal transduction related to transactivation of antioxidative genes regulated by ncRNAs. Nrf2 is a transcriptional factor that enters the nucleus in response to oxidative stress, resulting in increased expression of numerous neuroprotective genes. Nrf2 is regulated through the PI3K/Akt pathway, which plays key roles in regulating neuroprotection against oxidative stress. BDNF is a ligand of TrkB, which promotes neuronal survival and protects against apoptosis mediated through the PI3K/Akt pathway. BDNF can also bind to p75^NTR^—which was identified as a low-affinity nerve growth factor receptor—and BDNF can activate the NFκB pathway as well as the p38/JNK pathways. Boxes indicate miRNAs that target the signal transduction molecule, and the rounded rectangle indicates lncRNA.

**Figure 3 ijms-22-04245-f003:**
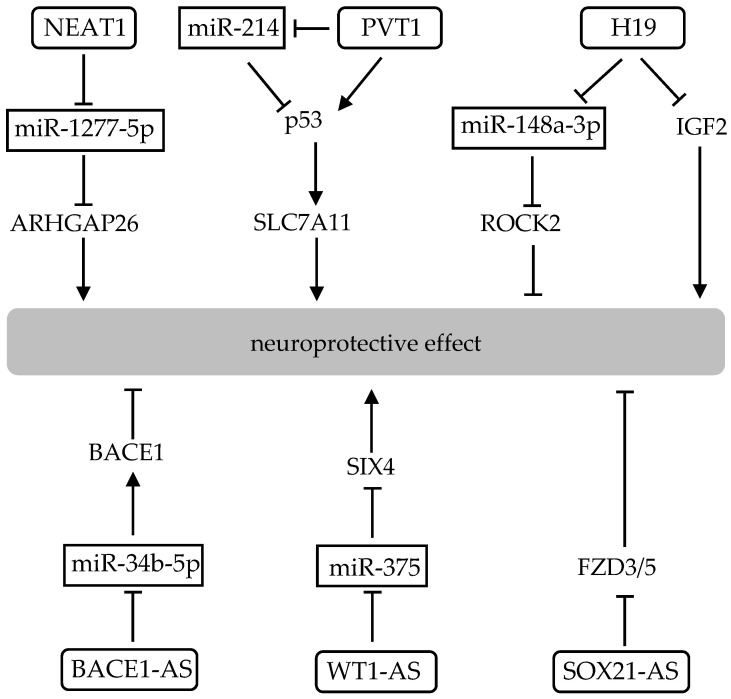
Interplay of lncRNAs and miRNAs in neuroprotective effect. LncRNAs play a key role in neuroprotection mainly by acting as a miRNA sponge. Boxes indicate miRNAs that target the signal transduction molecule, and the rounded rectangle indicates lncRNAs.

**Figure 4 ijms-22-04245-f004:**
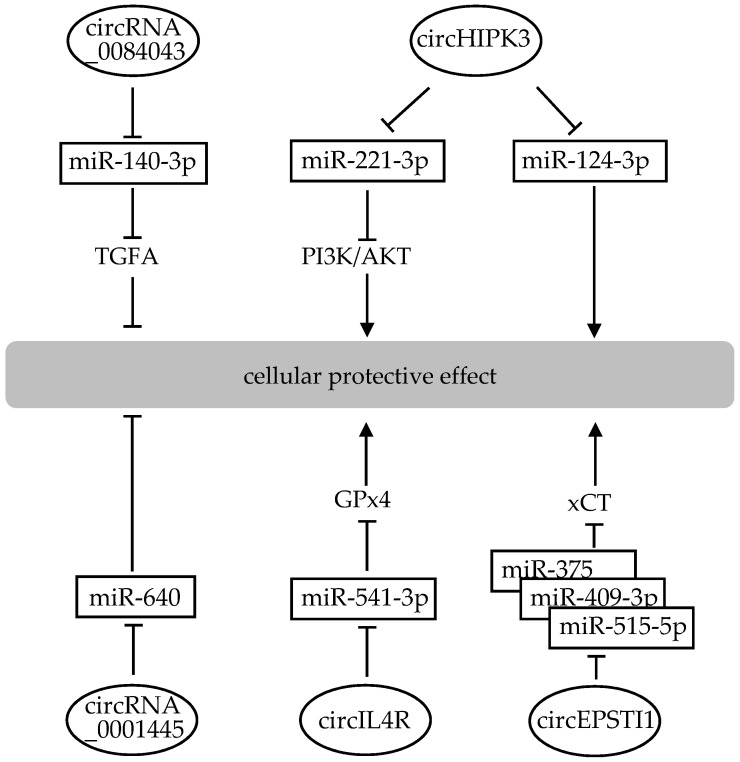
Interplay of circRNAs and miRNAs in cellular protective effect. CircRNAs play a key role in protective function mainly by acting as a miRNA sponge. Boxes indicate miRNAs that target the signal transduction molecule, and the oval indicates circRNAs.

**Table 1 ijms-22-04245-t001:** List of miRNAs regulated redox states in brain tissue or neuronal cells.

MiRNA	Direct Target	Related Pathway	Effect on Redox States			Brain Tissue or Neuronal Cell	Ref
			Glutathione	Antioxidative Enzymes	Oxidative Stress		
miR-7	Keap1	Nrf2 pathway	GSH ↑	GCLm ↑	CBA *^2^ ↓	SH-SY5Y cell	[103]
miR-23a-3p	Nrf2	Nrf2 pathway	n.d.	n.d.	ROS ↑ MDA ↑	brain *^1^	[101]
miR-25	KLF2	Nrf2 pathway	GSH ↓	GST ↓ Trx ↓	n.d.	hippocampus	[104]
miR-96-5p	EAAC1	Cys transport	GSH ↓	n.d.	ROS ↑	substantia nigra	[121]
miR-96-5p	NOVA1	Cys transport	GSH ↓	n.d.	ROS ↑	dentate gyrus of hippocampus	[122]
miR-98	HEY2	Notch signaling	GSH ↑	GPx ↑ SOD ↓	MDA ↓	hippocampus	[117]
miR-129-3p	MCU	MMP2 pathway	GSH/GSSG ↓	SOD ↓	ROS ↑	primary hippocampal neurons	[118]
miR-139	n.d.	Nrf2 pathway	GSH ↑	CAT ↑ SOD ↑	MDA ↓	SH-SY5Y cell	[107]
miR-144	n.d.	Nrf2 pathway	GSH ↓	GPx ↓	ROS ↑	SH-SY5Y cell	[108]
miR-146a	TRAF6	NF-κB pathway	n.d.	GPx ↑ SOD ↑	MDA ↓	brain *^1^	[116]
miR-153	Nrf2	Nrf2 pathway	n.d.	GCLc ↓	ROS ↑	SH-SY5Y cell	[102]
miR-199a-5p	MRP1	GSSG clearlance	GSSG ↑	n.d.	n.d.	primary cortical neurons	[123]
miR-200a	n.d.	PKA pathway	n.d.	GPx ↓ SOD ↓	MDA ↑	striatum	[119]
miR-200c-3p	RECK	PI3K/AKT pathway	n.d.	GPx ↓ SOD ↓	MDA ↑	hippocampus	[109]
miR-204-5p	BDNF	TrkB pathway	GSH ↓	SOD ↓	ROS ↑ MDA ↑	HT-22 cell	[110]
miR-214	PTEN	PI3K/AKT pathway	GSH ↑	SOD ↑	MDA ↓	SH-SY5Y cell	[112]
miR-320	Nox2	Nox2 pathway	n.d.	GPx ↑ CAT ↑ SOD ↑	ROS ↓ MDA ↓	primary neuron	[120]
miR-326	KLK7	p38/JNK pathway	n.d.	SOD ↓ GPx ↓	MDA ↑	striatum	[115]
miR-409	n.d.	PI3K/AKT pathway	GSH ↑	SOD ↑	ROS ↓	PC-12 cell	[111]
miR-410	TIMP2	p38/JNK pathway	n.d.	GPx ↑ SOD ↑	MDA ↓(serum)	hippocampal neurons	[113]
miR-486	NeuroD6	p38/JNK pathway	n.d.	GPx ↓	ROS ↑	spinal cord	[114]
miR-592	KIAA0319	Nrf2 pathway	GSH ↓	CAT ↓ SOD ↓	ROS ↑ MDA ↑	cortical astrocytes	[105]

Upward and downward arrows indicate increased and decreased level of redox markers, respectively. *^1^ Area of brain tissue were not specified in the article. *^2^ CBA: Coumarin boronate acid. n.d.; not detected.

**Table 2 ijms-22-04245-t002:** List of lncRNAs regulated redox states in brain tissue or neuronal cell.

LncRNA	Direct Target	Function	Effect on Redox States			Brain Tissue or Neuronal Cell	Ref
			Glutathione	Antioxidative Enzymes	Oxidative Stress		
AK046177	n.d.	Nrf2/CREB regulation acting with miR-134	n.d.	GPx ↓ SOD ↓	ROS ↑ MDA ↑	primary cortical cell	[127]
BACE1-AS	miR-34b-5p	BACE1 upregulation acting as miR-34b-5p sponge	n.d.	GPx ↓ SOD ↓	MDA ↑	substantia nigra	[131]
H19	miR-148a-3p	ROCK2 upregulation acting as miR-148a-3p sponge	n.d.	GPx ↓ SOD ↓	MDA ↑	Neuro2a cell	[129]
H19	IGF2	inhibition of antioxidative gene transcription	GSH ↓	GPx ↓ CAT ↓ SOD ↓	n.d.	hippocampal neuron	[130]
NEAT1	miR-1277-5p	ARHGAP26 upregulation acting as miR-1277-5p sponge	n.d.	GPx ↓ SOD ↓	MDA ↑	SK-N-SH cell	[125]
PVT1	miR-214-3p	TP53 and TFRC upregulation acting as miR-1277-5p sponge	GSH ↑	GPx ↑	MDA ↓	SK-N-SH cell	[126]
SOX21-AS1	FZD3/5	inactivation of Wnt signalin pathway	n.d.	GPx ↓ CAT ↓ SOD ↓	ROS ↑ MDA ↑ 4-HNE * ↑	hippocampal neuron	[133]
WT1-AS	miR-375	SIX4 upregulation acting as miR-375 sponge	n.d.	GPx ↑ SOD ↑	ROS ↓ MDA ↓	SH-SY5Y cell	[132]

Upward and downward arrows indicate increased and decreased level of redox markers, respectively. * 4-HNE: 4-hydroxy-2-nonenal. n.d.; not detected.

**Table 3 ijms-22-04245-t003:** List of circRNAs regulated redox states in tissue or cell.

CircRNA	Direct Target	Function	Effect on Redox States			Tissue or Cell	Ref
			Glutathione	Antioxidative Enzymes	Oxidative Stress		
circRNA_0084043	miR-140-3p	TGFA upregulation acting as miR-221-3p sponge	n.d.	GPx ↓ SOD ↓	MDA ↑	ARPE-19 cells	[136]
circHIPK	miR-221-3p	PI3K/AKT pathway activation acting as miR-140-3p sponge	n.d.	GPx ↑	MDA ↓	LECs	[137]
circHIPK	miR-124-3p	apoptosis induction acting as miR-124-3p sponge	n.d.	GPx ↓ SOD ↓	MDA ↑ LDH *^1^ ↑	HCM	[138]
circRNA_0001445	miR-640	protective function acting as miR-640 sponge	n.d.	GPx ↑ SOD ↑	MDA ↓	HUVECs	[139]
circIL4R	miR-541-3p	GPx4 upregulation acting as miR-541-3p sponge	n.d.	n.d.	ROS ↓ MDA ↓	hepatocellular carcinoma	[140]
circEPSTI1	miR-375 miR-409-3p miR-515-5p	SLC7A11 upregulation acting as sponge of miR-375, -409-3p and -515-5p	GSH/GSSG ↑	n.d.	Liperfluo *^2^ ↓	cervical cancer	[141]

Upward and downward arrows indicate increased and decreased level of redox markers, respectively. *^1^ LDH: lactate dehydrogenase, *^2^ Liperfluo: N-(4-Diphenylphosphinophenyl)-N′-(3,6,9,12-tetraoxatridecyl) perylene-3,4,9,10-tetracarboxydiimide. n.d.; not detected.

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
