# Peer review of "The Role of Non-Coding RNAs in the Neuroprotective Effects of Glutathione"

_ijms, 2021, doi:10.3390/ijms22084245_

Round 1

Reviewer 1 Report

In this review, the authors describe how the non-coding RNAs (including microRNA, long non-coding RNA and circular RNA) modulate the level of GSH and modify the oxidative stress states in the brain. The several types of non-coding RNAs are abundantly expressed in the brain, and their activation or inhibition could contribute to neuroprotection through the regulation of GSH synthesis and/or metabolism.

The publication would be appropriate.

Few minor comments:

 (1) To further clearly understand the paper for readers, a figure at section 4.2. Long non-coding RNAs would be desirable, likes to micro RNAs section.

 (2) Similarly, circular RNAs section also needs such as a figure.

Author Response

We are grateful for your comments. We took your comments into account in the revised version of manuscript as follows.

Comments:

Few minor comments:

 (1) To further clearly understand the paper for readers, a figure at section 4.2. Long non-coding RNAs would be desirable, likes to micro RNAs section.

 (2) Similarly, circular RNAs section also needs such as a figure.

Response :  In accord with your suggestions, we added figures for long non-coding RNAs (Figure 3) and circular RNAs (Figure 4).

Reviewer 2 Report

The review by 

The review by Kinoshita and Koji Aoyama aims to summarize the role of non-coding RNAs in the neuroprotective effects of 2 glutathione. This is a very specific and focused topic and for this reason the authors expanded their review on additional topics related to antioxidative defense of brain cells or neurons. As such, the title represents only a small part of the content and the text should be revised accordingly.

With the exception of some minor points (i.e. Figure 2 is missing the word “genes” after the “Transcriptional activation of antioxidative”) the abstract should be re-written to cover all the presented topics and the conclusions should be more precise and focused. The take-home message is unclear and the authors should consider to shorten the first general part of the manuscript and expand the part which presents the role of specific classes of non-coding RNAs. Overall, I feel that after extensive revision, the manuscript will appear more comprehensive and cohesive.    

Author Response

We are grateful for your comments and useful suggestions. As indicated in the responses that follow, we have taken all these comments and suggestions into account in the revised version of our paper.

Comment1: Figure 2 is missing the word “genes” after the “Transcriptional activation of antioxidative”

Response1: In accord with this suggestion, we added "genes" after the  “Transcriptional activation of antioxidative”

Comment2: the abstract should be re-written to cover all the presented topics

Response2:  In accord with this suggestion, we added sentences in the abstract (line 16-19, 24-25).

Comment3: the conclusions should be more precise and focused. The take-home message is unclear 

Response3:  In accord with this suggestion, we added sentences in the concolusion (line 948-951).

Comment4: the authors should consider to shorten the first general part of the manuscript 

Response4:  In accord with this suggestion, we deleted sentences in the manuscript (line 88, 95, 133, 159, 303, 305 and 307).

Comment5: the authors should expand the part which presents the role of specific classes of non-coding RNAs.

Response5:  In accord with this suggestion, we added sentences in the manuscript (line 339, 354 and 362).

Reviewer 3 Report

In this article, Kinoshita and Coll. review the role of non-coding RNAs in the neuroprotective effects of glutathione.

Especially, the Authors focus on the description of the antioxidant system and on the neuroprotective role of non-coding RNA.

In general, the review is interesting and well organized.

Just a minor point regarding the paragraph on circular RNAs: a graphical model on the predicted involvement of circRNAs in the antioxidant system could be appreciated.

Author Response

We are grateful for your comments. We took your comments into account in the revised version of manuscript as follows.

Comment: Just a minor point regarding the paragraph on circular RNAs: a graphical model on the predicted involvement of circRNAs in the antioxidant system could be appreciated.

Response: In accord with your suggestions, we added figure for circular RNAs (Figure 4).

Round 2

Reviewer 2 Report

Τha authors addressed all the points raised. The manuscript has been improved and is appropriate for final acceptance.